# The Development of Polylactic Acid/Multi-Wall Carbon Nanotubes/Polyethylene Glycol Scaffolds for Bone Tissue Regeneration Application

**DOI:** 10.3390/polym13111740

**Published:** 2021-05-26

**Authors:** Shih-Feng Wang, Yun-Chung Wu, Yu-Che Cheng, Wei-Wen Hu

**Affiliations:** 1Department of Urology, Cathay General Hospital, Taipei 10603, Taiwan; 8501008@gmail.com; 2School of Medicine, Fu-Jen Catholic University, New Taipei City 242062, Taiwan; 3Department of Chemical and Materials Engineering, National Central University, Zhongli District, Taoyuan City 32001, Taiwan; mrturtlewu@hotmail.com; 4Proteomics Laboratory, Department of Medical Research, Cathay General Hospital, Taipei 10630, Taiwan; 5Department of Biomedical Sciences and Engineering, National Central University, Zhongli District, Taoyuan City 32001, Taiwan

**Keywords:** electrospinning, dexamethasone, multi-wall carbon nanotubes, drug-loaded scaffolds, polylactic acid, polyethylene glycol, bone tissue engineering, osteogenic differentiation

## Abstract

Composite electrospun fibers were fabricated to develop drug loaded scaffolds to promote bone tissue regeneration. Multi-wall carbon nanotubes (MWCNTs) were incorporated to polylactic acid (PLA) to strengthen electrospun nanofibers. To modulate drug release behavior, different ratios of hydrophilic polyethylene glycol (PEG) were added to composite fibers. Glass transition temperature (*T*g) can be reduced by the incorporated PEG to enhance the ductility of the nanofibers. The SEM images and the MTT results demonstrated that composite fibers are suitable scaffolds for cell adhesion and proliferation. Dexamethasone (DEX), an osteogenic inducer, was loaded to PLA/MWCNT/PEG fibers. The surface element analysis performed by XPS showed that fluorine of DEX in pristine PLA fibers was much higher than those of the MWCNT-containing fibers, suggesting that the pristine PLA fibers mainly load DEX on their surfaces, whereas MWCNTs can adsorb DEX with evenly distribution in nanofibers. Drug release experiments demonstrated that the release profiles of DEX were manipulated by the ratio of PEG, and that the more PEG in the nanofibers, the faster DEX was released. When rat bone marrow stromal cells (rBMSCs) were seeded on these nanofibers, the Alizarin Red S staining and calcium quantification results demonstrated that loaded DEX were released to promote osteogenic differentiation of rBMSCs and facilitate mineralized tissue formation. These results indicated that the DEX-loaded PLA/MWCNT/PEG nanofibers not only enhanced mechanical strength, but also promoted osteogenesis of stem cells via the continuous release of DEX. The nanofibers should be a potential scaffold for bone tissue engineering application.

## 1. Introduction

Due to the limited sources of autologous and allogeneic bone tissues as well as the possibilities of pathogen infection and rejection during transplantation, synthetic scaffolds such as bone substitutes have received considerable attention to replace and treat large-sized bone defects. Appropriate physical, chemical, and biological cues can be provided by these scaffolds to guide osteoprogenitor cells differentiation and promote bone regeneration [1]. In order to provide a biomimetic environment, nanofibers are considered as excellent scaffolds because their architectural features are similar to the 3-D structure of the extracellular matrix (ECM), which is mainly composed of fibrillar collagen [2,3]. In addition, the high specific surface area of nanofibers can increase protein adsorption, including albumin, fibronectin, and laminin on their surfaces, and thus the adhesion and proliferation of cells can be promoted to accelerate tissue regeneration [4]. Their high porosity is also beneficial to the exchange of nutrients and wastes to facilitate cells ingrowth [5].

Polylactic acid (PLA) is a Food and Drug Administration (FDA)-approved synthetic biocompatible material and can be electrospun as nanofibers [6]. Although PLA nanofibers have been broadly utilized as scaffolds in tissue engineering, weak in mechanical strength restricts their application of healing bone defects. Furthermore, mechanical properties of scaffolds highly influence stem cells, which may govern their differentiation and commit them to specific lineages. Therefore, these nanofibers should be reinforced to allow their application to hard tissue. Multi-wall carbon nanotubes (MWCNTs) are commonly used as fillers to strengthen materials [7]. Electrospun nanofibers can be reinforced by MWCNTs incorporation [8]. In addition, MWCNTs are osteoconductive and have been proven to up-regulate genes involved in osteo-differentiation and promote the level of mineralization [9,10,11]. Therefore, nanofibers containing MWCNTs have been broadly investigated as scaffolds for bone tissue engineering applications [12,13,14].

In addition to mechanical support, scaffolds can also be applied as drug-carriers to regulate cellular physiology. Through the adjustment of scaffold composition, the delivery rate and duration can be delicately regulated to fulfill the therapeutic requirement during tissue regeneration. Dexamethasone (DEX) is a steroidal drug that is frequently used to reduce inflammatory response. Because DEX can regulate RUNX2, a critical transcription factor associated with osteogenesis, DEX accompanied with ascorbic acid and β-glycerophosphate is frequently used to guide bone marrow stromal cells to differentiate into osteoblasts and mineralization [15,16]. However, the administration of DEX in high concentration also leads to osteoporosis because over-dose DEX administration may shift the trend of mesenchymal stem cells differentiation toward adipocytes rather than osteoblasts [17]. Therefore, a sustainable low-dose delivery is highly required to reduce the side effect of DEX [18]. Using scaffolds to mediate drug delivery can control the drug distribution specifically in target site, suggesting that DEX delivery through scaffolds is a potential strategy to avoid systemic problem [19].

Poly (ethylene glycol) (PEG) is a hydrophilic polymer broadly used in pharmaceutical formation because it can facilitate drug-loading of hydrophobic carrier, such as PLA nanofibers [20]. Because PEG can be dissolved in aqueous environment, incorporation PEG in nanofibers can create channels to promote drug diffusion [21]. Therefore, in this study we fabricated MWCNT-containing PLA nanofibers as 3-D drug-loaded scaffolds. We hypothesized that the incorporated MWCNTs may improve the mechanical strength of nanofibrous scaffolds and promote osteogenesis. In addition, DEX was loaded to facilitate osteogenic differentiation. In order to increase the wettability of nanofibers PEG was added, and their concentrations can also manipulate the releasing rate and duration of DEX from nanofibers. Finally, rat bone marrow stromal cells (rBMSCs) were seeded into these nanofibers to investigate the potential of these scaffolds for bone tissue regeneration.

## 2. Materials and Methods 

### 2.1. Materials

Polylactic acid with molecular weight of 180~210 kDa is provided by Wei-Mon Industry Co. (Taichung, Taiwan). Multi-wall carbon nanotubes (MWCNTs) are acid-oxidated to the level of COOH of 1.8 mol.% (Legend Star International Co., New Taipei City, Taiwan), and their diameters and lengths are between 10–20 nm and 10–30 μm, respectively. Polyethylene glycol (PEG) with molecular weight of 8 kDa, dexamethasone, sodium dodecyl sulfate (SDS), and thiazolyl blue tetrazolium bromide (MTT), pluronic F-127 are obtained from Sigma-Aldrich (St. Louis, MO, USA). Dimethylformamide (DMF), dichloromethane (DCM), dimethyl sulfoxide (DMSO), Dulbecco’s modified Eagle medium (DMEM), and fetal bovine serum (FBS) are purchased from Thermo Fisher Scientific (Waltham, MA, USA).

### 2.2. Polymer Solution Preparation and Electrospinning Process

Acid-oxidated MWCNTs were dispersed in DMF. To increasing dispersion homogeneity, SDS and pluronic F-127 in 10 wt % of MWCNTs were added to MWCNT solution [22], which were ultrasonically treated for 3 h in cooling environment. Finally, the dispersive MWCNT solution was stirred at room temperature.

On the other hand, 10 wt % of PLA and different amount of PEG were dissolved in DCM at room temperature. Then, the dispersive MWCNTs were dropwisely added to the PLA/PEG solution under ultrasonication. The details of the amounts of MWCNT and PLA solutions for the electrospun solution preparations are listed in Appendix A. After stirring at room temperature for 1 day, the prepared solution can be applied for electrospinning. Regarding drug-loading fiber preparation, DEX was dissolved in DMF to prepared 4 wt % solution. Before electrospinning, 0.75 g of DEX solution was added to electrospun solution containing 1 g of PLA and was stirred for 20 min, so that the ratio of DEX to PLA would be 3 wt % in nanofibers.

### 2.3. Electrospinning Process

To electrospin nanofibers, the polymer solution was fed to 1 mL polypropylene syringe with needle of 19G needle tip that was connected to a power supply (Chargemaster CH50-P, Simco-ion, Hatfield, PA, USA). The working voltages of was 15 kV, and the feeding rate was 1 mL/h. To collect electrospun fibers, glass cover slips with diameters of 16 mm were warped on a grounded aluminum rotating mandrel that was rotated at 300 rpm. The working distances between the needle tip and the rotating mandrel collector was 12 cm. The electrospun nanofibers were dried in an oven for 1 day at 37 °C, and these prepared nanofibers would be applied as scaffolds for bone tissue engineering application.

### 2.4. Scanning Electron Microscopy (SEM)

Morphology of electrospun nanofibers was examined by scanning electron microscopy (SEM, 3500N, Hitachi, Tokyo, Japan), which were sputter-coated with gold before examination. The diameter distribution of nanofibers was calculated by 100 measurements based on SEM images. Regarding cell morphology, cells and nanofibers were fixed by glutaraldehyde and serially dehydrated by ethanol. Finally, specimens were lyophilized to completely remove moisture and gold sputtered before SEM analysis.

### 2.5. Tensile Test

The mechanical performance of the nanofibers was analyzed by a tensile test. Electrospun nanofibers were collected from the rotating mandrel, which were cut into rectangles with lengths of 3 cm and widths of 1 cm, where 3 cm was the side parallel to the circumference of the mandrel. These 3 cm fiber specimens were clamped on the tweezers of a universal testing machine (QC-513M1F, Cometech, Taichung, Taiwan) for 1 cm each end. Fibers were stretched in a constant speed of 3 mm/min. The displacement and corresponding pressure was continuously monitored until fracture [23].

### 2.6. Water Contact Angle Analysis

The hydrophobicity of the electrospun nanofibers was examined by a water contact angle analysis (Drop Shape Analysis system, DSA10, Kruss GmbH, Hamburg, Germany). Water drops were added to 4 points per sample, and their contact angles were presented as averages with standard deviations.

### 2.7. Chemical Characteristics of Nanofibers 

To determine the composition and functional groups of nanofibers, Fourier transform infrared (FT-IR) and X-ray photoelectron spectroscopy (XPS) were performed. After electrospinning for 2 h, the prepared nanofibers were adhered to the sample holder of FT-IR spectroscopy (FT/IR 410, JASCO, Tokyo, Japan). The measurement was in a resolution of 4 cm^−1^ between 4000 and 600 cm^−1^. Surface chemical analysis of nanofibers was carried out by XPS (K-Alpha, Thermo, Waltham, MA, USA). Peaks of C 1s and O 1s were resolvedly analyzed by iterative Gaussian/Lorentzian fitting (Magicplot, Saint Petersburg, Russia).

### 2.8. Differential Scanning Calorimetry (DSC)

In order to determine the response of nanofibers to temperature changes, DSC was applied for thermal analysis. We collected 6 mg of nanofibers and placed them in an aluminum pan. Then, the sample pans were evaluated by the DSC (Exstar6000 DSC, Seiko Instruments Inc., Chiba, Japan) to determine the difference in heat flow. The temperature was rise from 30 to 600 °C in a rate of 10 °C/min.

### 2.9. The Release of DEX from Nanofibers

The method of loading DEX to polymer solution for electrospinning was list in Section 2.2. These DEX-containing nanofibers were evaluated of their delivery by placing 0.1 mg of nanofibers in 4 mL of PBS at 37 °C. The released DEX was continuously sampled and measured by the absorbance at 242 nm, and the released DEX was quantified by comparing with the linear calibration results of standard solutions [24].

### 2.10. In Vitro Cell Culture of rBMSCs

Rat bone marrow stromal cells (rBMSCs) were harvested from 8-week-old Sprague–Dawley rats [25]. After being sacrificed, the femur was harvested and both sides were cut with forceps. A 10 mL syringe contained DPBS were used to wash out the cells. The rBMSCs were further separated from other cells such as red blood cells by FICOLL 400 (#F4375, Sigma-Aldrich/Merck). The purified rBMSCs were seeded in normal culture dish and maintained in regular medium of DMEM with 10% FBS. These cells were passaged 3–5 days in a subcultivation ratio of 1:3. The passage numbers of rBMSCs in this study were between 10–15. To evaluate the effect of DEX-containing nanofibers on osteogenesis, osteogenic medium was prepared by adding ascorbic acid-2-phosphate and 10 mM of β-glycerophosphate in DMEM, which were used during the osteogenesis experiment.

### 2.11. The Evaluation Cell Adhesion and Proliferation of Nanofibers

To evaluate whether cells may adhere and proliferate on nanofibers, an MTT test was performed. Nanofibers were electrospun on round coverslips in diameters of 1.6 cm, and then were placed in 24-well plates. After sterilizing DEX-containing nanofibers by UV treatment for 1 h, rBMSCs were seeded to nanofibers in density of 17,000 cells/cm^2^. All experiments were performed triplicate. After 1-, 3-, or 5-day culture, 100 μL of MTT solution (5 mg/mL in phosphate buffered saline (PBS)) and 900 μL of medium were added to wells and kept for 3 h at 37 °C. The supernatant was removed and 1 mL of DMSO was added to dissolve formazan, which then was spectrometrically analyzed at the wavelength of 550 nm.

### 2.12. Quantification of Calcium Deposition in Extracellular Matrix (ECM)

To determine the effect of DEX delivery on osteo-differentiation, rBMSCs were seeded on nanofibers that were electrospun on round coverslips in diameters of 1.6 cm, and then were placed in 24-well plates. All nanofibers containing DEX were examined. The seeding density of rBMSCs was 2.5 × 10^4^ cells/well. Osteogenic medium was applied in this experiment, which was changed every other day. Because osteoblasts differentiated from rBMSCs can deposit calcium to form mineralized tissue, the calcium deposition was qualitatively examined by Alizarin Red S staining and quantified by the calcium ortho–cresolphthalein complexone method (Ca-*o*-CPC) assay reaction after being treated in osteogenic medium for 14 and 21 days [26].

For Alizarin Red S staining, the cultures were rinsed with PBS and fixed by 1% glutaraldehyde in PBS for 30 min at 37 °C, and then stained by 2% for Alizarin Red S solutions for another 20 min at room temperature. After PBS rinse, the stained samples were observed by inverted microscope (Eclipse Ti-U, Nikon, Tokyo, Japan).

Regarding the Ca-*o*-CPC assay, the medium was removed and the wells were thoroughly rinsed with PBS. Then, 100 μL of 0.5 N acetic acid was added to release calcium ions. Equal volume (200 μL) of calcium binding reagent (1 g/L of *o*-cresolphthalein complexone, and 0.1 g/L of 8-hydroxyquinoline) and calcium buffer reagent (1.6 M of 2-amino-2-methyl-1-propanol, pH 10.7) were added to 10 μL of calcium released sample. After 15-min incubation at room temperature, 100 μL of purple-colored Ca-*o*-CPC complex was transferred to 96-well multiplates to optically measure its absorbance at wavelength of 575 nm [26,27]. The amount of calcium in the cell lysate was calculated according to the linear calibration results of the calcium chloride standard solutions.

### 2.13. Statistical Analysis

The statistical analyses were performed following a two-tailed Student’s *t*-test to make comparison and the errors were reported as standard deviations.

## 3. Results and Discussion

### 3.1. The Preparation of MWCNT-Containing PLA Nanofibers

Electrospun PLA nanofibrous scaffold is commonly used in tissue engineering because of its excellent biocompatibility, high specific surface area, and high porosity. However, its mechanical performance is relatively weak, which hinders its application on bone tissue engineering [28]. Therefore, MWCNTs were added to reinforce PLA nanofibers in this study. First, we prepared PLA containing MWCNTs without PEG to determine the optimal amount of MWCNTs.

Because solvents of the PLA solution, such as chloroform and DCM, are mostly highly volatile, they are difficult to utilize in an electrospinning process. Therefore, compatible solvent DMF is frequently used to avoid quick evaporation [29,30,31,32]. We dispersed MWCNTs to DMF and added to PLA solution to electrospin PLA nanofibers containing 0.1, 0.5, 1.25, and 3 wt % of MWCNTs, which were denoted as 0.1 C, 0.5 C, 1.25 C, and 3 C, respectively (Appendix A). The morphology of the prepared nanofibers was showed in Figure 1a. For the PLA fibers electrospun without MWCNTs, the diameters were wildly distributed between 200 and 1200 nm (661 ± 243 nm). In contrast, MWCNTs obviously reduced the diameters of the electrospun fibers and with the higher concentrations of MWCNTs, the fibers were thinner (Figure 1b). Furthermore, the diameters of the MWCNT-containing nanofibers were also evenly distributed compared with the pristine PLA fibers. Because the conductivity of polymer solution is improved by the added MWCNTs, high conductivity increases charges of polymer solution when the high voltage is applied to the tip of the needle. Therefore, the electrostatic repulsion of these charges is enhanced to counteract the surface tension to stretch the droplet on the tip, which thus decreases the size of the Taylor cone to shrink the diameters of electrospun nanofibers [22].

Figure 1c depicts the stress–strain curves of the tensile tests, and the mechanical performances are arranged in Table 1. Compared to the pristine PLA nanofibers, the nanofibers containing slight MWCNTs (i.e., 0.1 C and 0.5 C groups) demonstrated enhanced mechanical performances, including Young’s modulus, yield stress, and tensile strength. Because the MWCNTs own excellent mechanical strength, the nanofibers can be reinforced by their incorporation. On the other hand, MWCNTs increase entanglement to reduce slip within the polymer chains, and this slip inhibition also leads to the polymer chains being unable to rearrange to consume tension, so the elongation at the break is decreased. When we further increased MWCNTs to 1.25 and 3 wt %, the mechanical performance did not improve. We deduced that too many MWCNTs may aggregate to each other, and that their unevenly distribution may hamper mechanical performance [33]. Therefore, we applied 0.5 wt % MWCNTs in the following experiments.

### 3.2. Blending PEG to MWCNT-Containing PLA Fibers

Although MWCNTs can improve the mechanical performance of PLA fibers, the water contact angle of the PLA fibers increased from 132° to 135° when 0.5% MWCNTs were added, suggesting that both PLA and MWCNTs are hydrophobic materials, which may hinder drug delivery application of nanofibers (Figure 2a) [34]. To overcome this difficulty, the solution of the 0.5 C group was added with hydrophilic PEG was added to PLA so that the ratio of PEG to PLA in ratios of 0.1, 1, and 10 wt %, which were denoted as 0.5 C/0.1PEG, 0.5 C/1PEG, and 0.5 C/10PEG, respectively (Appendix A). The water contact angle results showed that PEG increased hydrophilicity of nanofibers that the water contact angle of 0.5 C/10PEG was reduced to 127°. These results suggested that the wettability of electrospun fibers can be improved by PEG incorporation, which should be beneficial to the encapsulation and delivery of DEX from nanofibers. Furthermore, the addition of PEG did not affect the morphology of nanofibers, so the sizes of the PEG-containing nanofibers were similar to those of the 0.5 C group (Figure 2b). 

We also applied FT-IR and XPS analysis to examine functional groups of nanofibers (Figure 3). For the FT-IR analysis, there is a specific peak of PEG at 1212.5 cm^−1^, which is correlated to the stretching vibration of C-O in the terminal of PEG chains (Figure 3a) [35]. This peak increased with the concentration of PEG in electrospun solution. The results of XPS spectrometry demonstrate a similar trend. The peaks of C 1s and O 1s highly depend on the composition of nanofibers (Figure 3b). Considering the structure of PLA and PEG, C 1s can be resolved into 4 peaks (Figure 3c) The peaks of 289.3 ± 0.1, 287.3 ± 0.1, and 285.3 ± 0.1 eV are carbons assigned to the –**C**OO, –**C**H–, and –**C**H_3_ of PLA, and the carbon of PEG contributed to the peak of 286.3 ± 0.1 eV [36]. Similarly, O 1s can be resolved into three peaks, i.e., 532.2 ± 0.2, 533.7 ± 0.2, and 532.8 ± 0.2 eV, which represent the carboxyl group (**O**=C–O–C) of PLA, the ester group (O=C–**O**–C) of PLA, and the alcohol (C–**O**–H) as well as ether (C–**O**–C) groups of PEG, respectively (Figure 3d) [37]. The area ratios of these resolved peaks are listed in Table 2 and Table 3. The nanofibers prepared by polymer solutions containing more PEG exhibited higher area ratios of 286.3 ± 0.1 eV and 532.8 ± 0.2 eV in C 1s and O 1s, respectively. Both FT-IR and XPS results suggest that PEG is successfully incorporated in nanofibers.
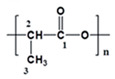


Regarding the effects of incorporated PEG on the mechanical performance of nanofibers, a tensile test was performed (Figure 4a). Table 4 lists the performance of nanofibers in tensile tests. The addition of PEG weakens nanofibers that Young’s modulus, yield stress, and tensile strength all decrease with the amount of PEG in nanofibers, however, their mechanical performances are still better than pristine PLA fibers. On the other hand, because the polymer chains of linear PEG are highly flexible [38], adding PEG to nanofibers can extend the elongation at break, especially that the elongation of the 0.5 C/10PEG Tgroup is almost 2.3 times that of the 0.5 C group. Although the result of the 0.5 C group showed that the added MWCNTs effectively reinforce PLA fibers, these MWCNT-containing nanofibers are rigid and may easily fracture. In contrast, the further addition of PEG not only preserves the reinforcement effects of MWCNTs but also increases the ductility of nanofibers, which should be beneficial to their clinical application.

Thermal transition of the nanofibers was examined by DSC analysis (Figure 4b), and their glass transition temperature (Tg), crystal temperature (Tc) and melting point (Tm) were determined through the DSC curves (Table 5). Because MWCNTs restrict the movement of PLA chains, the Tg of 0.5 C is slightly higher than that of the pristine PLA [39]. In addition, the proportionality of Tg of composite materials to their weight fraction can evaluate the compatibility of composite materials [40]. The glass transition temperatures (Tg) of PEG and PLA are −60 and 62.5 °C, respectively, so the Tg of the composite fibers decreased with increasing PEG ratios. Furthermore, their Tg are directly proportional to their weight ratios, suggesting that the added PEG can be evenly blended with PLA without phase separation. Crystal temperature (Tc) is the endothermic peak of DSC curve. Different studies indicated that the addition of nanomaterials serve as nucleated agents [41,42], and thus Tc of the 0.5 C group was lower than that of the pristine PLA fibers. Because PEG is a soft molecule, Tc of pure PEG is 40.5 °C, which is much lower than that of PLA [40]. The blended PEG may reduce the energy barrier to facilitate crystal formation, and thus the Tc decreases with increasing PEG. For the Tm results, MWCNTs slightly increase Tm due to their entanglement effect that higher energy is required to relax polymer chains. However, PEG addition does not significantly affect Tm.

### 3.3. Cell Adhesion and Proliferation on Nanofibers

In this study, MWCNT and PEG were incorporated to PLA to form composite nanofibers that can be applied as scaffolds to facilitate bone regeneration. To evaluate their potential in tissue engineering application, rBMSCs were seeded to nanofibers for 3 days and cell morphology was examined using SEM (Figure 5a). The seeded cells successfully adhered and spread on nanofibers, suggesting that these nanofibers provide an appropriate environment for cell growth. The low-magnification SEM (500×) photos show that cells on pristine PLA fibers are round and small. In contrast, cells grown on nanofibers containing MWCNTs exhibit an epidermal cell-like morphology. The high-magnification SEM photos (3000×) show that cells on nanofibers containing MWCNTs demonstrate rough surfaces with pseudopodia, whereas cells on pristine PLA fibers own smooth contour without pseudopodia. These results suggest that MWCNTs can increase focal adhesion through pseudopodia formation.

To investigate the viability of cells on nanofibers, an MTT assay was performed (Figure 5b). For the results of the first day, there was almost no difference between the groups, suggesting that the efficiencies of cell adhesion to these nanofibers are almost the same. Regarding the results of the 3rd and 5th days, the cells increased in all groups including the control groups of TCPS, suggesting that cell proliferation is successful in these nanofibers. The results of 0.5 C and pristine PLA are almost the same. In contrast, the cells on nanofibers that contained more PEG demonstrated lower MTT values, but the result of the 0.5 C/0.1PEG group can be maintained as that of the 0.5 C group, suggesting that slightly adding PEG does not affect cell proliferation. Because the hydrophilic property of PEG may reduce protein adsorption, the proliferation of cells grown on PEG containing nanofibers was probably thus affected [43]. However, PEG is a biocompatible material that is FDA approved for clinical use [44]; therefore, PEG should not cause cytotoxicity and should be applicable for tissue engineering.

### 3.4. Drug-Loaded Electrospun Nanofibers

In order to promote osteogenesis, DEX was loaded to nanofibers of pristine PLA, 0.5 C, 0.5 C/0.1PEG, 0.5 C/1PEG, and 0.5 C/10PEG, which are denoted as DL-PLA, DL-0.5 C, DL-0.5 C/0.1PEG, DL-0.5 C/1PEG, and DL-0.5 C/10PEG, respectively. These DEX loaded nanofibers were characterized by FT-IR analysis. The FT-IR spectra showed that there is a specific peak of DEX at 1600~1700 cm^−1^, which corresponded to the stretching vibration of the carbonyl group of DEX (Figure 6a) [45,46]. In addition, the nanofibers containing DEX all demonstrate this specific peak, suggesting that the DEX is successfully loaded to nanofibers (Figure 6b).

Because the DEX molecule contains the fluorine atom, XPS was applied to evaluate the loading efficiency of the DEX. The peak of F 1s is at 684–688 eV, which can be found in all DEX loaded nanofibers (Figure 6c,d). Therefore, the area of this specific peak is determined, which was normalized by the peak area of C 1s to determine the F/C molar ratio. The experimental and theoretical F/C molar ratios are list in Table 6.

Interestingly, the peak of F 1s of DL-PLA is larger than the other groups. The quantified results also indicate that the experimental F/C molar ratio of DL-PLA is higher than its theoretical value. In contrast, the experimental F/C molar ratios of nanofibers containing MWCNTs and PEG are all lower than the theoretical values (Table 6). Because the depth of XPS analysis is between 1 to 25 nm [47], DEX seems to distribute mainly on the surface of the DL-PLA nanofibers. On the other hand, Liao et al. have incorporated MWCNTs to PLA and polycaprolactone (PCL) for electrospinning [48]. Their results showed that MWCNTs tend to distribute inside nanofibers. Carbons of MWCNTs are sp2 hybridized, and thus their 2p orbitals extended from the planar sp2 can adsorb aromatic compounds, such as DEX, through π–π interaction [49]. Because the DEX are adsorbed by MWCNTs embedded in nanofibers, the experimental F/C molar ratios lower than the theoretical values.

### 3.5. Delivery of DEX from Nanofibers and Its Effects on Osteogenic Differentiation

Finally, we applied these DEX-containing fibers for bone regeneration application. To investigate the delivery profile, DEX-containing fibers were immersed in PBS at 37 °C, and the released DEX was quantified by its absorbance at 242 nm (Figure 7a). The releases of DEX from DL-0.5 C fibers are all lower than those from DL-PLA fibers in all sampling times, suggesting that DEX release is hindered due to its adsorption to MWCNTs, which is in agreement to the XPS results. On the other hand, the incorporation of PEG can promote DEX release. The DL-0.5 C/10PEG group demonstrated a burst release of 25% of the loaded DEX in the first 9 h, but the following release was much low, even surpassed by the other groups. Different from the DL-0.5 C/10PEG, both DL-0.5 C/1PEG and DL-0.5 C/0.1PEG exhibit continuous release of DEX, and the more PEG, the faster DEX release.

Because PEG is extremely hydrophilic, these PEG chains tend to migrate toward the surfaces of nanofibers when the fibers are immersed in aqueous environment [50], we assume that nanofiber structure may be loosened during PEG migration to cause burst release. However, when these migrated PEG molecules completely cover the surfaces of nanofibers, a dense layer of PEG is formed to eventually retard the release of hydrophobic DEX. To prove our hypothesis, we examined water contact angles of DEX-containing nanofibers before (day 0) and after immersed in PBS for 4 days (Figure 7b). Water contact angles of all nanofibers were all decreased after 4 days in PBS because of the ingress of water to polymers. This reduction was not significant in the DL-PLA and DL-0.5 C groups. In contrast, the results of PEG-containing nanofibers all significantly increased their hydrophilicity that the higher PEG, the lower water contact angles, especially the DL-0.5 C/10PEG group that demonstrated a dramatically reduction of water contact angles from 127.8° ± 2.8° to 53.4° ± 7.3°. These results suggested that PEG migration may affect the hydrophobicity of nanofibers to regulate the release of DEX, and the surfaces of DL-0.5 C/10PEG became highly hydrophilic after immersing in aqueous environment, which thus retarded the release of DEX.

Then, we seeded rBMSCs to DEX loaded nanofibers and cultured them in medium containing ascorbic acid and β-glycerophosphate to evaluate the promotion extent in osteogenic differentiation ability of these nanofibers. The differentiated osteoblasts from rBMSCs can deposit calcium in the extracellular matrix (ECM) as a process called mineralization [51]. The formed mineralized tissue was stained by Alizarin Red S (Figure 7c) and the deposited calcium was quantified by a colorimetric assay of Ca-*o*-CPC complex method (Figure 7d). When rBMSCs were seeded on fibers for 14 days, there was almost not significant Alizarin Red S staining in the DL-PLA, DL-0.5 C and DL-0.5 C/10PEG groups. However, the bone-like nodules and obvious staining were found in the DL-0.5 C/0.1PEG and DL-0.5 C/1PEG groups, suggesting that these rBMSCs on these fibers began to deposit mineral in ECM [52]. In the late stage (day 21), all nanofibers except the DL-0.5 C/10PEG group successfully led rBMSCs to form mineralized tissue. The quantification results also indicated the same trends. The performances of the DL-PLA and DL-0.5 C groups were almost the same, no matter day 14 or day 21, which were lower than those of the DL-0.5 C/0.1PEG and DL-0.5 C/1PEG groups, and the DL-0.5 C/1PEG group caused the highest calcium deposition. Interestingly, cells grown on DL-0.5 C/10PEG did not increase calcium deposition on day 21, suggesting that DL-0.5 C/10PEG fibers were incapable of inducing osteogenic differentiation.

Actually, we also have seeded rBMSCs on nanofibers without DEX loading (i.e., PLA, 0.5 C, 0.5 C/0.1PEG, 0.5 C/1PEG, and 0.5 C/10PEG nanofibers) and mineralization and calcium deposition cannot be found in these groups, suggesting that DEX delivery is essential to trigger rBMSCs osteo-differentiation (data not shown). The quantification results of mineralization indicated that differentiation of rBMSCs highly depends on the delivery profiles of DEX from nanofibers. Because DEX can control the activity of RUNX 2, which is a key transcription factor associated with osteogenesis [15,16], the concentration of DEX has to be maintained in the early stage of differentiation. In the first week, the releases of DEX from DL-PLA and DL-0.5 C groups were almost the same and the incorporation of PEG increased DEX release that the more PEG, the higher DEX delivery. These release trends explain the corresponding results of calcium deposition. Regarding the DL-0.5 C/10PEG group, DEX can only be quickly released in the first 9 h. Because no more DEX was further released and the medium was changed every other day in this study, cells on DL-0.5 C/10PEG fibers can only be treated by DEX in the first 2 days. Therefore, DL-0.5 C/10PEG fibers are not suitable to facilitate bone tissue formation due to their burst release of DEX without sustainability.

## 4. Conclusions

In this study, we successfully electrospun MWCNT/PEG/PLA composite nanofibers. These nanofibers were reinforced by MWCNTs, and the incorporation of PEG improved the ductility of nanofibers to avoid brittle break. In addition, these MWCNTs can adsorb DEX to facilitate the even distribution of DEX in the nanofibers. The duration and releasing rates of DEX release can be manipulated by the amount of PEG in nanofibers, i.e., as the concentration of PEG in nanofiber increased, the release of DEX was faster. Finally, the differentiation of rBMSCs on these fibers demonstrated that continuous release of DEX from the DL-0.5 C/0.1PEG and DL-0.5/1PEG groups effectively accelerated osteogenic differentiation and improved mineralization, suggesting the potential of these nanofibrous scaffolds for bone tissue engineering application.

## Figures and Tables

**Figure 1 polymers-13-01740-f001:**
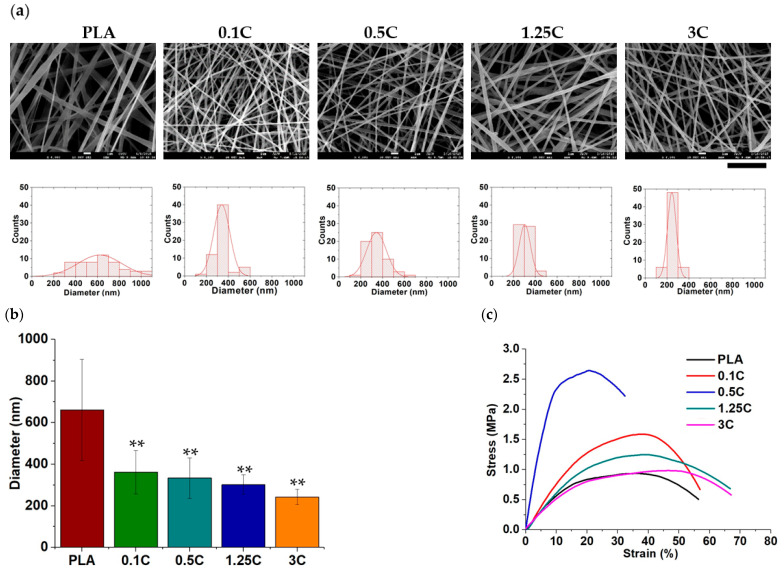
Characteristics of MWCNT-containing PLA nanofibers. (**a**) The morphology and size distribution of nanofibers were examined by SEM analysis. (scale bar = 5 µm); (**b**) the average size of MWCNT-containing nanofibers. (**: *p* < 0.01 compared to the pristine PLA group); (**c**) mechanical performances of nanofibers were evaluated by a tensile test.

**Figure 2 polymers-13-01740-f002:**
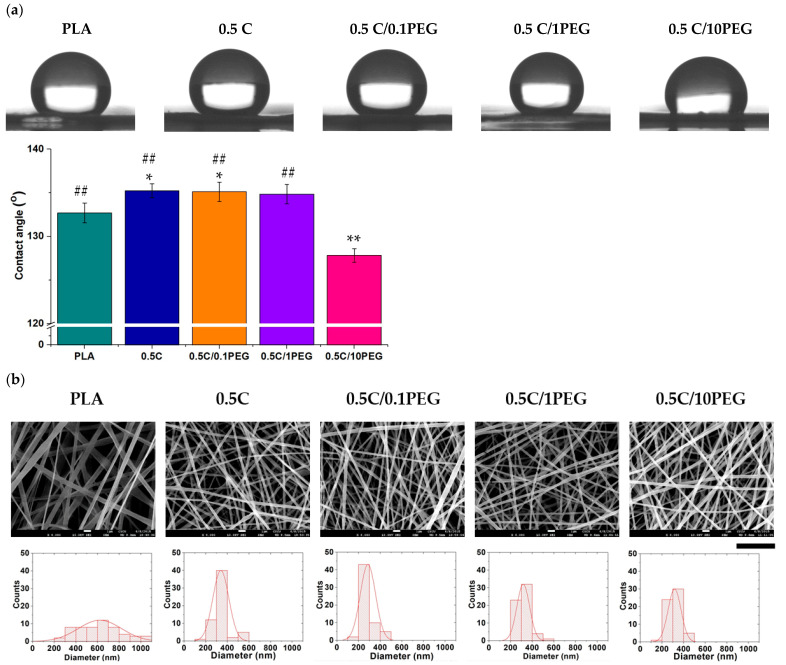
Characteristics of PEG-containing nanofibers: (**a**) the hydrophobicity of the nanofibers was evaluated by water contact analysis. (*: *p* < 0.05, **: *p* < 0.01 compared to the pristine PLA group; ##: *p* < 0.01 compared to the 0.5 C/10PEG group) (**b**) The morphology and size distribution of nanofibers were examined by SEM analysis. (scale bar = 5 µm) (**: *p* < 0.01 compared to the pristine PLA group).

**Figure 3 polymers-13-01740-f003:**
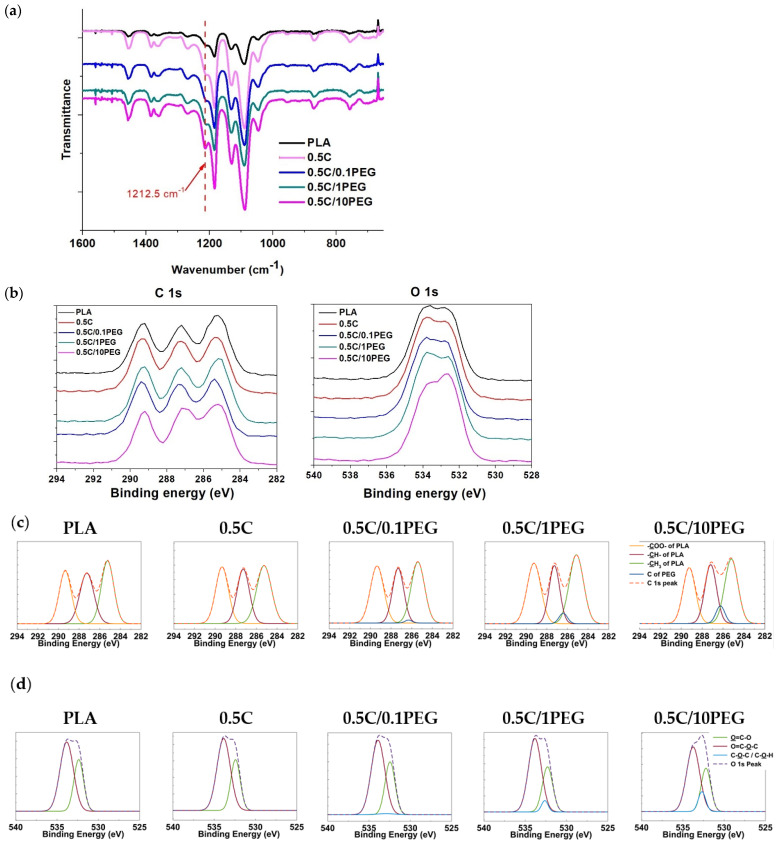
(**a**) The FT-IR spectra of nanofibers. The adsorption of 1212.5 cm^−1^ is the specific peak PEG due to its stretching vibration of C–O in the terminal of PEG chains. (**b**) The XPS spectra of nanofibers. The C 1s peak and O 1s peak are at 284~291 eV and 532~536 eV, respectively. (**c**) The C 1s (dashed line) can be resolved into 4 peaks, and (**d**) the O 1s (dashed line) are resolved into 3 peaks, which can be applied to distinguish the PLA and PEG (blue lines) components.

**Figure 4 polymers-13-01740-f004:**
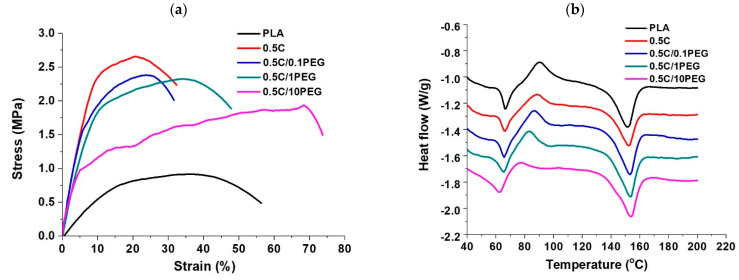
Characteristics of PEG-containing nanofibers by (**a**) tensile test and (**b**) DSC analysis.

**Figure 5 polymers-13-01740-f005:**
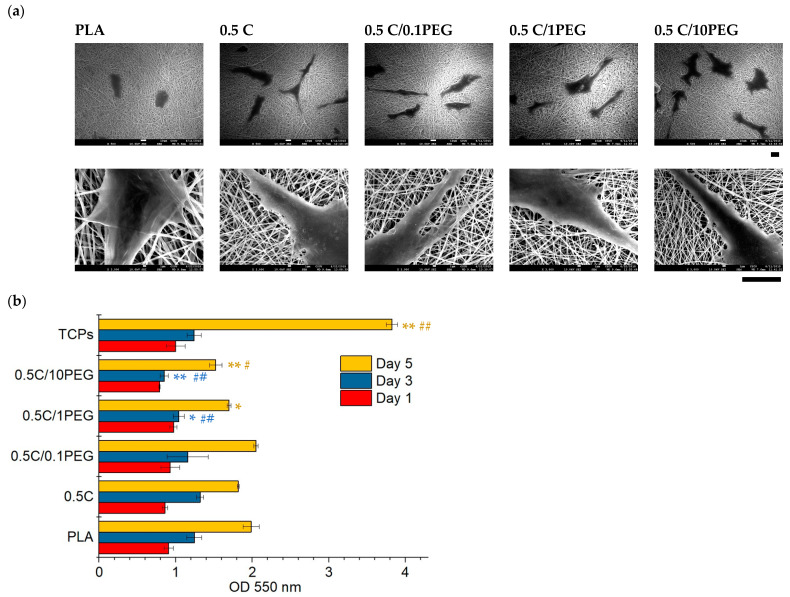
Cell adhesion and proliferation on nanofibers. (**a**) The rBMSCs were seeded to nanofibers for 3 days, and their morphology was examined by SEM. The top and bottom are low-magnification and high-magnification images, respectively. (scale bars = 10 µm). (**b**) The MTT results of cells grown on nanofibers for 1, 3, and 5 days. (*: *p* < 0.05, **: *p* < 0.01 compared to the pristine PLA group; #: *p* < 0.05, ##: *p* < 0.01 compared to the 0.5 C group).

**Figure 6 polymers-13-01740-f006:**
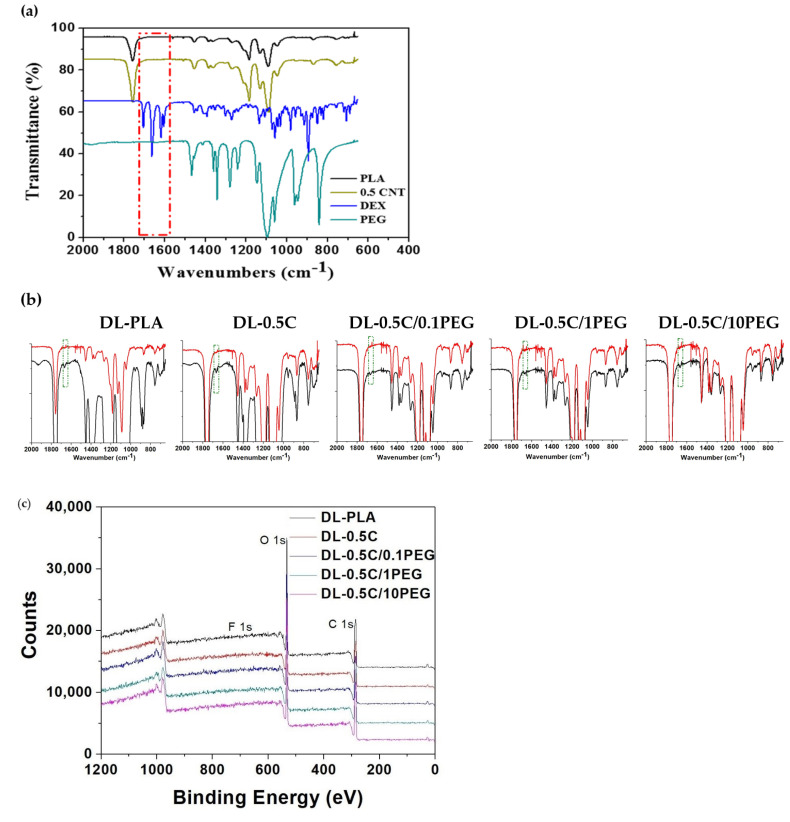
FT-IR and XPS analysis of DEX-loaded nanofibers. (**a**) The FT-IR spectra of PLA nanofibers 0.5 C nanofibers, DEX, and PEG. These spectra showed that only DEX demonstrates peaks between 1600 and 1700 cm^−1^. (**b**) The FT-IR spectra of nanofiber before (red) and after (black) DEX loading. The green dotted line indicated the adsorption region between 1600 and 1700 cm^−1^. (**c**) The XPS spectra were applied to investigate elements of nanofibers. (**d**) The F 1s peaks of nanofibers before (black) and after (red) DEX loading were investigated. Green dashed lines were baselines for the calculation of the area of F 1s peaks.

**Figure 7 polymers-13-01740-f007:**
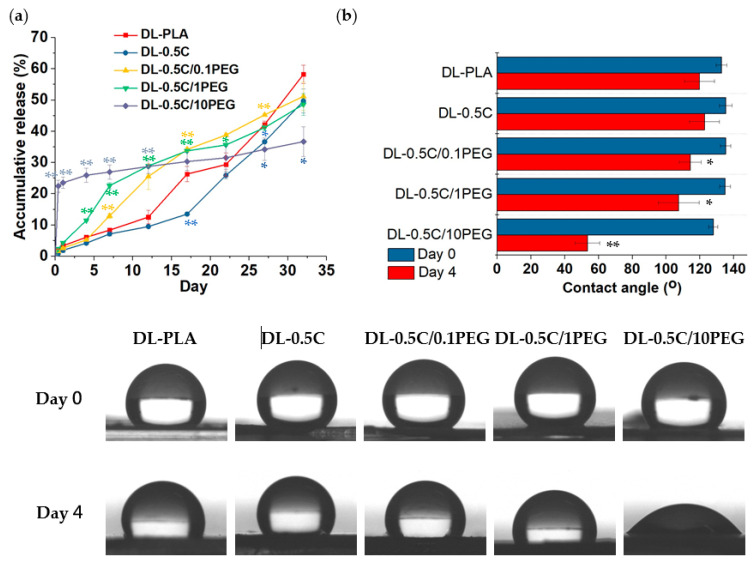
(**a**) The accumulative release of DEX from nanofibers was evaluated by incubating in PBS at 37 °C. (*: *p* < 0.05, **: *p* < 0.01 compared to the DL-PLA group). (**b**) Water contact angles of DEX-loading nanofibers were evaluated immediately and placing in PBS for 4 days. (*: *p* < 0.05, **: *p* < 0.01 compared to the results of day 0). (**c**) The effect of nanofibers on promoting osteogenesis was evaluated by seeding rBMSCs for 14 and 21 days, and the mineralized tissues were stained by Alizarin Red S. (scale bar = 500 µm). (**d**) Calcium deposition by the differentiated rBMSCs was analyzed by the Ca-*o*-CPC complex method. (**: *p* < 0.01 compared to the DL-PLA group).

**Table 1 polymers-13-01740-t001:** Mechanical performances of MWCNT-containing nanofibers.

	PLA	0.1 C	0.5 C	1.25 C	3 C
Young’s modulus (MPa)	5.390	7.120	24.30	5.570	4.450
Tensile strength (MPa)	0.936	1.587	2.657	1.249	0.984
Elongation at break (%)	56.33	56.87	32.35	66.74	67.04
Yield stress (MPa)	0.674	1.175	2.276	0.958	0.690

**Table 2 polymers-13-01740-t002:** Area ratios of characteristic peaks in C 1s spectra.

	–COO of PLA (C_1_) (289.3 ± 0.1 eV)	–CH– of PLA (C_2_) (287.3 ± 0.1 eV)	–CH_3_ of PLA (C_3_) (285.3 ± 0.1V)	O–C–C of PEG (286.3 ± 0.1 eV)
PLA	34.3%	25.4%	40.4%	0%
0.5 C	32.1%	29.8%	38.1%	0%
0.5 C/0.1PEG	34.4%	27.9%	36.8%	0.9%
0.5 C/1PEG	33.1%	24.5%	39.0%	3.5%
0.5 C/10PEG	28.9%	28.8%	35.8%	6.5%

**Table 3 polymers-13-01740-t003:** Area ratios of characteristic peaks in O 1s spectra.

	O=C–O–C of PLA (532.2 ± 0.2 eV)	O=C–O–C of PLA (533.7 ± 0.2 eV)	C–O–C or C–O–H of PEG (532.8 ± 0.2 eV)
PLA	34.2%	65.8%	0%
0.5 C	33.2%	66.8%	0%
0.5 C/0.1PEG	33.9%	64.9%	1.2%
0.5 C/1PEG	31.3%	63.7%	5.0%
0.5 C/10PEG	29.6%	59.9%	10.5%

**Table 4 polymers-13-01740-t004:** Mechanical performance of PEG-containing nanofibers.

	PLA	0.5 C	0.5 C/0.1PEG	0.5 C/1PEG	0.5 C/10PEG
Young’s modulus (MPa)	5.390	24.30	26.10	19.30	18.70
Tensile strength (MPa)	0.936	2.657	2.391	2.367	1.937
Elongation at break (%)	56.33	32.35	31.60	47.79	73.71
Yield stress (MPa)	0.674	2.276	1.610	1.750	0.976

**Table 5 polymers-13-01740-t005:** Tg, Tc, and Tm of nanofibers determined by DSC analysis.

(°C)	PLA	0.5 C	0.5 C/0.1PEG	0.5 C/1PEG	0.5 C/10PEG
Tg	62.5	63.0	61.5	60.0	50.5
Tc	90.5	88.8	86.5	83.2	77.3
Tm	151.3	153.0	153.3	153.3	153.2

**Table 6 polymers-13-01740-t006:** Area ratios of F 1s/C 1s.

Area Ratio of F 1s/C 1s	DL-PLA	DL-0.5 C	DL-0.5 C/0.1PEG	DL-0.5 C/1PEG	DL-0.5 C/10PEG
Theoretical ratios	0.18%	0.17%	0.17%	0.17%	0.16%
Experimental ratios	0.23%	0.12%	011%	0.10%	0.10%

## Data Availability

None.

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
