# Peer review of "The Development of Polylactic Acid/Multi-Wall Carbon Nanotubes/Polyethylene Glycol Scaffolds for Bone Tissue Regeneration Application"

_polymers, 2021, doi:10.3390/polym13111740_

Round 1

Reviewer 1 Report

In this manuscript of “The Development of Polylactic Acid/Multi-Wall Carbon Nanotubes/Polyethylene Glycol Scaffolds for Bone Tissue Regeneration Application”, Composite electrospun fibers were fabricated to develop drug loaded scaffolds to promote bone tissue regeneration. Although the present study was not in-depth, the whole scheme was designed all-roundly. However, I am puzzled by somewhere in the manuscript. The following are the questions and comments of this manuscript:

  1. The authors should address the existence questions of the current study in the Introduction part.
  2. Some experimental methods seem to lack references, there was none references in section 2.1 to 2.10.
  3. In line 251-252, “the solution of the 0.5C group was added with hydrophilic PEG in concentrations of 0.1, 1, and 10 wt.%”, why was there such a big difference in wt. % selection?
  4. In Table 6, why did F/C molar ratio of DL-PLA higher than its theoretical value ?
  5. The conclusion part did not make a good summary of the results of the research, so it is suggested to improve it.
  6. The format of the references should be unified and in line with the journal standards. In this manuscript some journal titles were abbreviated, some were the full name, it is suggested to carefully check and modify.

After going through the manuscript, I feel that authors need to revise the paper to reach the level of Polymers.

Reviewer 2 Report

It's a distinct idea to combine Polylactic acid, Multi-wall carbon nanotubes and polyethylene glycol to fabricate scaffolds for bone regeneration.  The authors have done detailed study. However, there should be a great effort needed to improve the methodology and results sections. So many discrepancies in the results section. 
Abstract
composite fibers all demonstrated good biocompatibility: mention by which experiment, this was proven?
The XPS results: Explain
osteogenic differentiation of rBMSCs and facilitate mineralized tissue formation: Mention How this was evaluated?
 but also continuously released DEX and promote osteogenesis of stem cells.: Change to " but also promote osteogenesis of stem cells by continuous release of DEX."
Did you investigate the separate role of PLA/MWCNT/PEG nanofibers without DEX on osteogenesis?
 the more PEG in nanofibers, the faster DEX released: It is recommended a slow-drug release for effective  therapeutic treatment. Because if the drug released so faster, it may have some side-effects, usually prolonged release may have good outcome in therapeutic treatment. Based on the above concept, the addition of PEG might minimise the efficiency of nanofibers? How do you explain?
1. Introduction
 are similar to the 3-D structure of extracellular matrix (ECM) which is mainly composed of fibrillar collagen: The present study uses PLA/MWCNT/PEG nanofibers, if so how this statement relates to this study?
DEX is frequently used to..differentiate into osteoblasts and mineralization [11]: With inducers or without? The reference No 11 used here was review work, not the original research paper to prove the effect. Provide an original research paper. Refer:Hong D et al. Osteoblastogenic effects of dexamethasone through upregulation of TAZ expression in rat mesenchymal stem cells. J Steroid Biochem Mol Biol. 2009, 116: 86-92. 10.1016/j.jsbmb.2009.05.007.
DEX in high concentration also leads to osteoporosis : The faster release of DEX leads to osteoporosis, Therefore the system developed by authors with more PEG in nanofibers may cause osteoporosis due to the faster DEX release?
MWCNTs are osteoconductive ... promote the level of mineralization.[9,10]: These reference work did not properly claim the evidance, insufficient works. The Reference no 9 by Chakraborty et al. describes the proliferation and  morphology by SEM and fluorescence of MG 63 cell seeded on MWCNTs cylindrical tubes and In vitro test in SBF is not solid study to confirm the stimulatory osteogenesis. The cell staining for osteogenesis such as calcium, alkaline phosphatse and mRNA expression were not done. So the first study is incomplete study. And the second study by Martinelli  et al.: Very simple study reported just cytotoxicity of MG-63 and mRNA expression (again so solid evidance like mineral staining and protein expression). This study did also not conduct the proper experiments, it may increase the proliferation of osteoblasts cells but osteogenesis is different. You have to understand whether treating your materials with the precursors such as mesenchymal stem cells or stromal cells can induce osteoblasts lineage cells formation without any inducers.
DEX administration may shift the trend of mesenchymal stem cells differentiation toward adipocytes rather than osteoblasts: So what is the appropriate concentration (recommended dosage) of DEX for osteoblast differentiation?
a sustainable low-dose delivery is highly required to reduce the side effect of DEX: In this case, what could be the optimum concentration of PEG in nanofibers, that is suitable for low-dose delivery of DEX? 
Discuss the previous studies reporting  the scaffold fabrication with MWCNT-containing PLA nanofibers especially for bone growth. Few reference works are listed below for you.
A Hudecki et al. Composite nanofibers containing multiwall carbon nanotubes as biodegradable membranes in reconstructive medicine, Nanomaterials, 2019
B Huang et al.  Fabrication and characterisation of 3D printed MWCNT composite porous scaffolds for bone regeneration
BK Shrestha  et al. Bio-inspired hybrid scaffold of zinc oxide-functionalized multi-wall carbon nanotubes reinforced polyurethane nanofibers for bone tissue engineering. Authors are free to choose some other important articles related to their field. 
2. Materials and Methods  
 different amount of PEG were dissolved: Give the specific dose amount of PEG. Total volume of mixture.
dispersive MWCNTs were dropwisely added to PLA/PEG solution: Give specific ratio of MWCNTs and PLA/PEG, total volume.
 The electrospun nanofibers were dried: Explain clearly how many types of nanofibers prepared totally and composition? From the manuscript title, this manuscript is dealing with Scaffolds for Bone Tissue Regeneration, but the authors stated nanofibers not scaffolds, they are not same. 
Regarding drug-loading fiber preparation, DEX was dissolved in DMF: Explain the concentration of each sample in reaction mixture and mixing ratio.
collected from rotating mandrel were collected. : Not a correct sentence.
 fiber specimens were fixed with 1 cm: Not a correct sentence.
 the strain rate was 3 mm/min. : After this explain how it was done.
Rat bone marrow stromal cells (rBMSCs) were harvested from 8-week-old Sprague-Dawley rats. : Explain the detailed procedure of rBMSCs harvesting from rats in a separate section.
These cells were maintained in DMEM with 10% FBS.: How the cells were passaged? Maximum passages used in experiments?
osteogenic supplement: Bought or maked by authors? Company details?
was applied to medium during the osteogenesis experiment.: used osteogenic medium or DMEM medium?
 After sterilization by UV treatment for 1 h, rBMSCs were: This means, you sterilize rBMSCs for 1 h and seeded? Write properly. Also explain how many groups and group details.
The MTT assay is used to measure cellular metabolic activity as an indicator of cell viability and proliferation. For biocompatibility, the immune cells need to be cultured on scaffolds.
For Alizarin Red S staining, the cultures: Explain how the cells were cultured? group details? Control group?
Regarding the Ca-o-CPC assay: Again same comment as above. What was the control group?
wavelength of 575 nm. : Reference?
3. Results and Discussion
The results and discussion: The effect of PEG on PLA nanofibers were completely missing in this section. But stated in method section. Refer Line 101: 10 wt% of PLA and different amount of PEG were dissolved in DCM
 MWCNTs are added in this study. : Re-frame
 MWCNTs to DMF and added to PLA solution to electrospin PLA nanofibers containing: What about PEG? It’s missing here. 
PLA nanofibers containing 0.1, 0.5, 1.25, and 3 wt.% of MWCNTs: Mention these details in method section. Also on what basis the concentration was selected?
PLA nanofibers containing 0.1, 0.5, 1.25, and 3 wt.% of MWCNTs: What about PEG? Refer Line 101: 10 wt% of PLA and different amount of PEG were dissolved in DCM
the fibers are thinner (Figure 1b).: Use past tense in writing your results.
distributed compared with pristine PLA fibers.: Where do you present the results of pristine PLA fibers?
conductivity of polymer solution is improved by the added MWCNTs: Where it is presented?
Figure 1. Characteristics of MWCNT-containing PLA nanofibers: It’s not PLA, it should be PLA-PEG nanofibers? Keep consistent phrases throughout MS. Change in all Figures as well. 
FIgure 1 & Table 1: Provide the results of PLA nanofibers prepared with different concentrations of PEG.
Figure 2: Here it is confusing. Fig1 shows PLA nanofibers but Fig.2 shows PEG nanofibers. PEG-containing nanofibers or MWCNT-containing PLA nanofibers.? Image 2b shows the SEM images of PLA nanofibers,  0.5 wt.% MWCNT nanofiber and 0.5C/PEGs nanofibers. Explain how many types of nanofibers were prepared totally? How did you prepare 0.5 wt.% MWCNT nanofiber alone? Provide method information. 
 132° to 135° when CNTs were added: CNTs or MWCNT? Keep consistent wording.
yield stress, and tensile strength all decrease with the amount of PEG in nanofibers: So adding PEG decreases the mechanical properties, in this case, do you claim PEG has potential in improving mechanical properties of nanofibers?
 however, their mechanical performances are still better : What do you mean? How it is better since PEG decreased mechanical properties? 
Based on the tensile strength and YM, incorporation of 0.1,1 and 10 PEG in nanofibers is not recommended for in vitro cell cultures. 
which were denoted as 0.5C/0.1PEG, 0.5C/1PEG, and 0.5C/10PEG: What about PLA? Not added in these nanofibers?
Figure 3.: Axis label is missing in all the images?
 cell proliferation is successful in these scaffolds. : Scaffolds or nanofibers? Define properly.
cells increased in all groups: Including control group or except?
more PEG demonstrated lower MTT values: This is the drawback of the present study. After seeing data, the increasing PEG downregulated the cell proliferation of BMSCs cells, that means PEG has cytotoxicity on bone cells growth and should not supposed to use in bone tissue engineering. 
In each figure and table, explain the abbreviation of each group such as 0.5C,  0.5C/0.1 PEG, 0.5C/1 PEG and 0.5C/10 PEG details. 
All tables and Figures: Statistical data SD and p value are missing. Provide the statistics wherever applicable.
Figure 5 Biocompatibility of nanofibers: The data confirmed that the authors used the dosage more than LD50 value so got lower cell number for higher dose. It is recommended to use the concentration lesser than 0.1 PEG and test cell viability.
DEX was loaded in weight ratio of 3% to nanofibers of pristine PLA, 0.5C, 0.5C/0.1PEG, 0.5C/1PEG, and 0.5C/10PEG: Why did you use 1 and 10 PEG? Because, these two concentrations were potentially decreased the cell viability.
DL-0.5C/0.1PEG, DL-0.5C/1PEG, and DL-0.5C/10PEG: so PLA was not incorporated?
Figure 6: Properly label the axis in each figure.
 released DEX was quantified by its absorbance at 242 nm: Explain in method section
than those from DL-PLA fibers: Is this control group?
 (b) Water contact angles of DL-0.5C/10PEG were evaluated immediately: DL-PLA, DL-0.5C/0.1 PEG and DL-1 PEG?
incorporation of PEG can promote DEX release: But the faster the drug release is not recommended, slow release is much better
hydrophilic when they were immersed in PBS for 4 days,: This might be due to evaporation of water molecules.
these DL-0.5C/10PEG nanofibers were examined by a water contact angle analysis: For complete analysis, use other samples also comparatively. 
more PEG, the faster DEX release. : Is this desirable property?
ascorbic acid and β-glycerophosphate to evaluate the promotion extent in osteogenic differentiation ability of these nanofibers. : The osteogenic ability of these nanofibers should be confirmed without inducers, use culture medium alone. 
to evaluate the promotion extent in osteogenic differentiation ability of these nanofibers.: Provide the detailed procedures osteogenic differentiation. How long the cells were cultured? 
obvious staining were found in the DL-0.5C/0.1PEG and DL-0.5C/1PEG groups.: What was the reason behind this variation? Why higher concentration 10 PEG did not observe bone-like nodules?
all nanofibers except the DL-0.5C/10PEG group: Why?
 DL-PLA and DL-0.5C groups were almost the same, no matter Day 14 or Day 21, which were lower than those of the DL-0.5C/0.1PEG and DL-0.5C/1PEG groups: Why DL-0.5C group has lower than the DL-0.5C/0.1PEG and DL-0.5C/1PEG groups
DL-0.5C/10PEG fibers were incapable of inducing osteogenic differentiation: This need to be tested without osteogenic inducers.
DEX can control the activity of RUNX 2 : Where is the evidence? Provide data in this work?
The discussion part is very poor and not solid prove or supporting evidence for each property studied. Refer some more work, and explain properly the reason for each result with proper evidence.
4. Conclusion
the more PEG in nanofibers, the faster DEX released. : Repeating the same.

Round 2

Reviewer 1 Report

The suggestions were considered by the authors and contributed further to scientific quality. Therefore, the manuscript was revised adequately and can be accepted for publication in Polymers.

Reviewer 2 Report

The authors have done extensive revision and it's really satisfactory. Pleased to accept this manuscript in its present form.